# Oscillation Characteristics of an Artificial Cochlear Sensory Epithelium Optimized for a Micrometer-Scale Curved Structure

**DOI:** 10.3390/mi13050768

**Published:** 2022-05-13

**Authors:** Hiroki Yamazaki, Yutaro Kohno, Satoyuki Kawano

**Affiliations:** Graduate School of Engineering Science, Osaka University, Toyonaka 560-8531, Osaka, Japan; yamazaki@me.es.osaka-u.ac.jp (H.Y.); boc01326@gmail.com (Y.K.)

**Keywords:** sensorineural hearing loss, cochlear implant, poly(vinylidene fluoride trifluoroethylene), fully implantable device, frequency selectivity

## Abstract

Based on the modern microelectromechanical systems technology, we present a revolutionary miniaturized artificial cochlear sensory epithelium for future implantation tests on guinea pigs. The device was curved to fit the spiral structure of the cochlea and miniaturized to a maximum dimension of <1 mm to be implanted in the cochlea. First, the effect of the curved configuration on the oscillation characteristics of a trapezoidal membrane was evaluated using the relatively larger devices, which had a trapezoidal and a comparable curved shape designed for high-precision in vitro measurements. Both experimental and numerical analyses were used to determine the resonance frequencies and positions, and multiple oscillation modes were clearly observed. Because the maximum oscillation amplitude positions, i.e., the resonance positions, differed depending on the resonance frequencies in both trapezoidal and curved membrane devices, the sound frequency was determined based on the resonance position, thus reproducing the frequency selectivity of the basilar membrane in the organ of Corti. Furthermore, the resonance frequencies and positions of these two devices with different configurations were determined to be quantitatively consistent and similar in terms of mechanical dynamics. This result shows that despite a curved angle of 50–60°, the effect of the curved shape on oscillation characteristics was negligible. Second, the nanometer-scale oscillation of the miniaturized device was successfully measured, and the local resonance frequency in air was varied from 157 to 277 kHz using an experimental system that could measure the amplitude distribution in a two-dimensional (2D) plane with a high accuracy and reproducibility at a high speed. The miniaturized device developed in this study was shown to have frequency selectivity, and when the device was implanted in the cochlea, it was expected to discriminate frequencies in the same manner as the basilar membrane in the biological system. This study established methods for fabricating and evaluating the miniaturized device, and the proposed miniaturized device in a curved shape demonstrated the feasibility of next-generation cochlear implants.

## 1. Introduction

Sound is perceived by humans by converting sound vibrations into neural signals in the cochlea. The cochlea has a two-and-a-half turn spiral structure with a base radius of 7 mm and a height of 5 mm [1]. In this study, the basilar membrane and the organ of Corti are considered as the core structures that mechanically and electrically recognize sound, and we focus on their biological functions. When the cochlear spiral is uncoiled and straightened for simple understanding, the basilar membrane would appear as a trapezoidal membrane that varies in width from 0.1 to 0.5 mm over its length of 30 mm. Because of the spatially inhomogeneous stiffness of the basilar membrane, the position at low stiffness resonates with lower frequencies and the position at high stiffness resonates with higher frequencies. Thus, the basilar membrane differentiates sound frequencies based on resonance positions, which is known as the frequency selectivity function. Two types of hair cells are aligned along the side edge of the basilar membrane and respond differently to the basilar membrane oscillations. The oscillations of the basilar membrane are controlled by the active expansion and contraction of the outer hair cells’ body. The dynamic range and frequency selectivity of hearing in human are considerably improved because of the cells’ somatic electromotility, allowing them to discriminate the sound pressures of 0–120 dB and frequencies of 20 Hz–20 kHz [2,3]. When the basilar membrane oscillates in response to external sound stimuli, the inner hair cells produce neurotransmitters, which are transmitted to the brain as nerve signals, allowing us to recognize sound. However, the loss of function in hair cell can result in sensorineural hearing loss, which is one of the causes of deafness and the inability to detect sound. Damaged hair cells never regenerate in humans [4], and cochlear implants, which are extensively commercialized, are the most effective treatment for sensorineural hearing loss [5,6]. The cochlear implant comprises both external and internal devices. The external device captures sounds and converts them into electrical signals, which are discharged through electrodes implanted in the cochlea to electrically activate nerve cells rather than the damaged hair cells. Although cochlear implants are recognized as the best clinical solution to sensorineural hearing loss in the current technology and their designs and performance have been improved since their inception [7,8], there are certain inconveniences because of the presence of external devices [9]. Typically, patients remove batteries during sleeping and bathing, and their excessive energy consumption necessitates frequent charging. Furthermore, the expense of cochlear implantation is extremely high and some people prefer not using a cochlear implant. Recently, there has been considerable interest in a fully insertable cochlear implant as the future technology that does not require any extracorporeal devices, and advances in fabrication technologies using microelectromechanical systems (MEMS) have made it possible to fabricate transducers on a micrometer scale.

Many research groups have developed various prototypes for fully insertable cochlear implants that do not require any external apparatus. The concept of developing such implants is based on biomimetics, with the key functions being the frequency selectivity of the basilar membrane and the conversion of sound/electrical signals of the inner hair cells. Commonly used methods for reproducing the frequency selectivity include the fabrication of trapezoidal membranes [10,11,12,13,14,15,16,17,18] and beams of different lengths [19,20,21,22,23,24]. Organic piezoelectric materials made of polyvinylidene difluoride (PVDF) [10,11,13,17,18,24] or poly(vinylidene fluoride-co-trifluoroethylene) (PVDF-TrFE) [25] and inorganic materials comprising lead zirconate titanate [12,23], zinc oxide [14], and aluminum nitride [19,20,21] have been used for sound/electrical signal conversion. We and several other research groups have attempted to stimulate the auditory nerve using the prototypes when the device was placed outside the guinea pig’s body, and electrodes were inserted into the cochlea. Auditory brainstem responses were successfully measured, and the feasibility of a fully insertable cochlear implant was confirmed for the first time [26,27,28]. However, two commonly known bottlenecks in the development of fully insertable cochlear implants are (1) the lack of power available to stimulate the nerve using biocompatible materials and (2) the difficulty of achieving a reproducible frequency range within the dimensions of the cochlea. We [17,18,29] and other groups [30,31,32,33] addressed the former issue by developing a nonlinear feedback control system that mimics the amplification of outer hair cells. However, there have been few studies of microdevices being implanted in the cochlea, and their performance, particularly in terms of mechanical dynamics, has yet to be thoroughly established. In this study, a novel curved and miniaturized artificial cochlear sensory epithelium that can be implanted in the cochlea was fabricated, and the oscillation characteristics were analyzed using experimental and numerical methods. The in vitro prototypes first investigated the difference in the resonance characteristics of trapezoidal and curved membrane devices with the set boundary conditions of the synthetic basilar membrane. The prototype device was oscillated at multiple frequencies in the experiment, and the oscillation amplitude at each position in the trapezoidal and curved membrane was measured to analyze local resonance frequencies and positions. Furthermore, natural frequencies and eigenmodes were numerically obtained using the finite element method. The resonance characteristics of both trapezoidal and curved membrane devices were reported to be quantitatively consistent, and the effect of the designs on implanting the device in the cochlea was negligible, as anticipated by the simple Euler–Bernoulli beam theory. Second, a miniaturized device, which can be implanted into the cochlea, was fabricated using MEMS technologies for in vivo experiments in future. The oscillation characteristics of the miniaturized device comprising an organic piezoelectric film P(VDF-TrFE) were measured; additionally, eigenfrequencies and modes were evaluated. The local resonance frequencies in air experiments and numerical analysis were 157–277 and 103–178 kHz, respectively, and the resonance positions obtained in the experimental and numerical results were in quantitatively good agreement. One of the possibilities of a fully insertable cochlear implant is frequency discrimination in a manner identical to the basilar membrane, and the frequency response of the proposed device was quantitatively validated.

## 2. Theoretical and Experimental Methods

### 2.1. Prototype of the Artificial Cochlear Sensory Epithelium and Experimental Setup for High-Throughput Oscillation Measurements

To examine the effects of curving an artificial cochlear sensory epithelium, the oscillation characteristics of a trapezoidal membrane device, which has been examined in previous studies [10,17,18], and a curved one were compared. Therefore, the trapezoidal and curved membrane prototypes for in vitro experiments were fabricated in this study. An acrylic plate with a trapezoidal or curved shape hole was molded using a three-dimensional (3D) printer (Objet Eden 260S, Stratasys, Eden Prairie, MN, USA). A piezoelectric membrane made of PVDF was used for the prototypes. As the compound membrane, Al thin-film electrodes were deposited on both sides of the PVDF film (KF piezo film, Kureha, Tokyo, Japan) with a thickness of 110 μm. The artificial cochlear sensory epithelium shown in Figure 1 was fabricated by sandwiching the compound membrane between the two acrylic plates with the trapezoidal or curved shape hole and constructing a fixed boundary at the edge of the hole. It is important to thoroughly determine the boundary conditions for uniquely identifying the vibrate mode in a continuum, and we confirmed that the fixed boundary conditions are satisfied using extensive trial and error. The *x* and *y* axes of the trapezoidal membrane device are defined as the trapezoid’s longitudinal and orthogonal directions, respectively. In the curved membrane device, curvilinear coordinates (ξ, η) are used, indicating that the ξ-axis is defined as the arc with a radius of 30 mm and an angle of 60° and the η-axis as the direction perpendicular to it. The trapezoidal membrane has a length of 31.4 mm in the *x* direction and has the widths in the *y* direction of 2 and 4 mm at *x* = 0 and 31.4 mm, respectively. The curved membrane has a length of 31.4 mm in the ξ direction and has the widths in the η direction of 2 and 4 mm at ξ = 0 and 31.4 mm, respectively. In the research field of the hearing system, the present device is a well-used model in which the cochlea is unrolled and straightened, and multiple prototypes with a similar shape have been developed as a first step toward the development of fully insertable cochlear implants.

In the experiment with the in vitro devices, an AC voltage $V=V_{\mathrm{in}}\sin\left(2\pi f_{\mathrm{in}}t\right)$ amplified 10 times via a high-speed bipolar power supply (HSA4014, NF Corp., Kanagawa, Japan) was applied to the devices through the deposited electrodes. The inverse piezoelectric effect induced the oscillation of the device, thereby preventing the effects of oscillation of the acrylic plates and the stage on which the device was mounted and enabling a highly accurate measurement with an accuracy of <5%. A laser Doppler vibrometer (LDV; AT0023/AT3600, Graphtec Corp., Tokyo, Japan) was used to measure the device’s oscillation while an automatic stage (SGSP26-150/SGSP20-35, Sigma Koki Co., Ltd., Tokyo, Japan) was moved to scan the measurement position. A data acquisition (DAQ) system was used to control the experimental apparatus and analyze the results. This DAQ system comprised a controller (NI PXIe-1082, NI PXIe-8840, National Instruments, Austin, TX, USA), a multichannel analog output module (NI PXIe-6738, National Instruments, Austin, TX, USA) for electrical stimulation, and a vibration module (NI PXIe-4492, National Instruments, Austin, TX, USA) for high-throughput data analysis. The input frequency $f_{\mathrm{in}}$ was varied from 8 to 13 kHz in 0.1 kHz steps, and the amplitude of the displacement in the *z*-axis was analyzed to measure the natural frequency and oscillation modes. The sampling rate and number of samples were set to 200 kHz and 200, respectively. The frequency bands below 0.5 kHz and above 20 kHz, which are outside the measuring range, were cut using the LDV’s high- and low-pass filters.

### 2.2. Fabrication of the Miniaturized Artificial Cochlear Sensory Epithelium Based on MEMS Technologies

Figure 2a shows the schematic of the miniaturized artificial cochlear sensory epithelium for in vivo experiments using guinea pigs. In the future, the device will be installed in the cochlea as an alternative to the organ of Corti: the frequency of a sound is specified based on the spatial position of the resonance and the auditory nerve is electrically stimulated using the potential difference generated using the piezoelectric effect. The device’s dimensions were optimized and miniaturized to the point where it could be implanted into the animal’s cochlea, as shown in Figure 2b. The width of the outer frame was 500 μm, which is sufficiently narrow to fit in the cochlear duct. The device was curved to fit the biological cochlea, which has a spiral structure. Although the cochlea has two-and-a-half turns (900°), this device was designed for 50° as a first step, and the angle of turn will be increased in the future. In the miniaturized device, curvilinear coordinates (ξ, η) are used, indicating that the ξ-axis is defined as the arc with a radius of 1.15 mm and an angle of 50°, and the η-axis is defined as the direction perpendicular to it. The miniaturized device has a length of 803 μm in the ξ direction and the widths in the η direction of 100 and 300 μm at ξ = 0 and 803 μm, respectively. As shown in Figure 2c, the device had a stacked structure. An elastic layer was installed in the miniaturized device such that the neutral axis of the film is shifted to provide strong piezoelectric signals, which are used to stimulate the auditory nerve. A piezoelectric film installed on the miniaturized device was made of PVDF-TrFE. It has been shown that PVDF-TrFE is more suitable for transducers than PVDF because the electromechanical coupling factor $k_{31}$ of PVDF-TrFE ($k_{31}$ = 0.3 [34]) is higher than that of PVDF ($k_{31}$ = 0.14 [35]), and the mechanical $\tan\delta_{m}$ and dielectric loss tangent $\tan\delta_{e}$ of PVDF-TrFE ($\tan\delta_{m}$ = 0.0335 and $\tan\delta_{e}$ = 0.089 [36]) are lower than those of PVDF ($\tan\delta_{m}$ = 0.0789 and $\tan\delta_{e}$ = 0.184 [36]). Three pairs of upper electrodes are used to obtain electrical signals based on biological tonotopy and control the oscillation through feedback control [10,17,18,29]. Because the electrodes are metallic and the piezoelectric film and elastic layer are organic materials, the adhesion of these materials is difficult. However, this is achieved by introducing Ti as an adhesive layer. To develop an artificial cochlear sensory epithelium for next-generation cochlear implants, it is necessary to verify the frequency selectivity and piezoelectricity. In this study, only the capacitance of the miniaturized device was successfully measured as an electrical evaluation (0.463 pF), and this result was almost the same value derived from the physical properties (0.433 pF) [37]. On the other hand, the dimensions of the electrodes attached to the micrometer-scale device were 0.008 mm${}^{2}$, and the method of connecting the device to the electrical measurement setup has not been established. In our previous studies [17,18,24], voltages of a few mV could be measured, but the device is miniaturized to the micrometer-scale, and the magnitude of the voltage will decrease on the micro-volt, making it difficult to obtain reproducible data. In this study, we focused on investigating the resonance characteristics for accumulating basic features of the miniaturized device as a first step, and the piezoelectricity of the miniaturized device must be measured in the future.

The fabrication process of the miniaturized artificial cochlear sensory epithelium is described as follows and summarized in Figure 3. The device developed in this study was fabricated from resin materials except for the electrodes, whereas a previous study [26] used deep-etching processes for preparing substrates; therefore, the device will be fabricated by injection molding in the future, and stable mass production is expected.
(i)A glass substrate (C024321, Matsunami Glass Ind., Ltd., Osaka, Japan) was ultrasonically cleaned using isopropanol.(ii)A sacrificial layer of Cr:Au:Cr = 70:50:70 nm was sputtered on the glass substrate.(iii)A negative photoresist of SU-8 (3005, Kayaku Advanced Materials, Inc., Westborough, MA, USA) for preparing an elastic layer was diluted with the same weight of cyclopentanone and uniformly coated on the surface of the sacrificial layer for 30 s using a spin coater at a rotation speed of 6000 rpm (MS-B100, Mikasa Co., Ltd., Tokyo, Japan). Setting the thickness of SU-8 as 1 μm, the glass substrate was baked on a hotplate at 65 °C for 1 min and at 95 °C for 6 min. Subsequently, a design of the elastic layer was printed on the SU-8 surface using a maskless ultraviolet (UV) exposure equipment (μMLA, Heidelberg Instruments Mikrotechnik GmbH, Heidelberg, Germany), and the glass was baked again at 65 °C for 1 min and at 95 °C for 6 min. The glass substrate was immersed in a developer (SU-8 Developer, Kayaku Advanced Materials, Inc., Westborough, MA, USA) for 90 s and rinsed using isopropanol. Finally, the glass substrate was baked at 150 °C for 5 min.(iv)A positive photoresist (AZ5214E, Merck, Darmstadt, Germany) is spin-coated on SU-8 for 30 s at 4000 rpm. A design of the upper electrodes was printed on the photoresist surface after prebaking for 1 min at 100 °C. After a reversal bake for 3 min at 120 °C, the entire surface of the glass substrate was exposed to UV light. The exposed photoresist was developed by immersing the membrane in developer solution (NMD-W 2.38%, TOKYO OHKA KOGYO Co., Ltd., Tokyo, Japan) for 90 min and rinsing with ultrapure water. To remove the photoresist, the glass substrate was immersed in ethanol for 2 min. Upper electrodes were prepared employing Au with a thickness of 50 nm and Ti with a thickness of 10 nm as an adhesion layer using the sputter-deposition technique. Patterned electrodes were obtained using the lift-off process with N,N-dimethylformamide.(v)To exclude contaminants and moisture, a P(VDF-TrFE) solution was prepared in a glove box. P(VDF/TrFE) powder (75/25 mol%) (FC25, Piezotech, Pierre-Benite, France) was dried at 130 °C for >12 h. The powder was dissolved in a diethyl carbonate solvent for >12 h at 130 °C. After setting the thickness of P(VDF-TrFE) film as 1 μm via spin coating at 1000 rpm for 30 s twice, the glass was baked on a hotplate at 130 °C for 3 min.(vi)After coating the P(VDF-TrFE) layer, the Ti/Au/Ti layer was sputtered to act as bottom electrodes.(vii)The glass was baked at 130 °C for 2 h for the crystallization of the P(VDF-TrFE) film. During polarization processing, a DC voltage of 100 V was applied to the P(VDF-TrFE) layer for 5 min.(viii)A negative photoresist of SU-8 (3050, Kayaku Advanced Materials, Inc., Westborough, MA, USA) was uniformly coated for 30 s on the substrate at a rotation speed of 800 rpm. Setting the thickness of SU-8 as 200 μm, the glass substrate was baked at 65 °C for 3 min and at 95 °C for 1 h. Subsequently, a design for the substrate layer was printed on the SU-8 surface using maskless UV exposure equipment, and the glass substrate was baked at 65 °C for 1 min and at 95 °C for 10 min. The glass substrate was immersed in the developer for 15 min and rinsed using isopropanol. Finally, the glass was baked for 5 min at 150 °C.(viii)A Cr etchant was used to remove the miniaturized artificial cochlear sensory epithelium from the glass substrate.

### 2.3. Numerical Analysis

PVDF or PVDF-TrFE were used as the material for preparing an artificial cochlear sensory epithelium with trapezoidal and curved membranes because of its biocompatibility (its toxicity tests have been demonstrated in our previous study [25]). PVDF or PVDF-TrFE are designated as a linear elastic material, and the numerical analysis of the oscillation characteristics for the experimental devices employs the following equation [38]: (1)$$\begin{array}{c} \rho \frac{\partial^{2}\zeta_{x}}{\partial t^{2}}=\frac{\partial \sigma_{x}}{\partial x}+\frac{\partial \tau_{yx}}{\partial y}+\frac{\partial \tau_{zx}}{\partial z},\;\;\rho \frac{\partial^{2}\zeta_{y}}{\partial t^{2}}=\frac{\partial \tau_{xy}}{\partial x}+\frac{\partial \sigma_{y}}{\partial y}+\frac{\partial \tau_{zy}}{\partial y},\;\;\rho \frac{\partial^{2}\zeta_{z}}{\partial t^{2}}=\frac{\partial \tau_{xz}}{\partial x}+\frac{\partial \tau_{yz}}{\partial y}+\frac{\partial \sigma_{z}}{\partial z}, \end{array}$$
where ρ is the density of the elastic body, $\zeta_{m}\left(x,y,z,t\right)$
(m=x,y,z) is the displacement in the *m*-axis), $\sigma_{m}\left(x,y,z,t\right)$ is the tensile stress in each principal axis, and $\tau_{mn}\left(x,y,z,t\right)$({m,n}={x,y,z}) is the shear stress on the mn-plane. For isotropic media, $\sigma_{m}$ and $\tau_{mn}$ are given by
(2)$$\begin{array}{cc} \; & \sigma_{m}=E_{mm}\varepsilon_{mm}+\sum_{n\neq m}E_{nn}\varepsilon_{nn},\;\;\;\tau_{mn}=E_{mn}\varepsilon_{mn}\;\mathrm{for}\;n\neq m,\\ \; & E_{mm}=\frac{1-\nu }{\left(1+\nu )(1-2\nu \right)}E,\;\;\;E_{nn}=\frac{\nu }{\left(1+\nu )(1-2\nu \right)}E\;\mathrm{for}\;m\neq n,\;\;\;E_{mn}=\frac{E}{2(1+\nu )}\;\mathrm{for}\;n\neq m, \end{array}$$
where $\varepsilon_{mn}$ is the strain tensor and *E* and ν are Young’s modulus and Poisson’s ratio, respectively, which are unique for isotropic materials. However, it is difficult to determine the elastic modulus *E* of the PVDF bimorph membrane, which varies depending on the material processing. Therefore, the *E* of the prototype devices was determined under the condition that the first-order natural frequencies obtained in the experimental results should match with the frequencies obtained via numerical analysis. The *E* of the miniaturized device was set to 11 GPa based on previous studies [39,40]. The ν of the prototype and miniaturized devices was set to 0.35 based on a previous study [37]. The four sides of the device were set as fixed bounds to keep the boundary conditions consistent with experiments. COMSOL Multiphysics 5.5 (COMSOL, Inc., Stockholm, Sweden) was used to investigate the natural frequencies and eigenmodes of each device employing the Arnoldi method.

## 3. Results and Discussion

### 3.1. Numerical and Experimental Results of In Vitro Prototypes Devices

Figure 4 shows the natural frequencies $f_{r,\mathrm{num}}^{\left(i\right)}$ and corresponding oscillation modes *i* (*i* = 1, 2, 3) of a numerically analyzed artificial cochlear sensory epithelium with a trapezoidal and curved membrane. For each frequency, the colormap shows the amplitude of displacement in the *z*-axis normalized by the maximum value in the xy-plane. In numerical analysis, the dimensions of the prototype with a trapezoidal and curved membrane were determined based on the experimental device, as shown in Figure 1. The results demonstrated that the oscillation amplitude was symmetrical in relation to the *x*- and ξ-axis and the oscillation mode was fundamental in the *y* direction. Furthermore, as $f_{r,\mathrm{num}}^{\left(i\right)}$ increased, several antinodes with a number equal to *i* were observed and the positions of the maximum amplitude shown by the red color decreased along the *x*- and ξ-axis. As shown in Table 1, the positions of maximum amplitude are defined as resonance positions $x_{r,\mathrm{num}}^{\left(i\right)}$ and $\xi_{r,\mathrm{num}}^{\left(i\right)}$ for the trapezoidal and curved membrane devices, respectively. Therefore, the oscillation characteristics of both of the devices represented numerically by $f_{r,\mathrm{num}}^{\left(i\right)}$ and $x_{r,\mathrm{num}}^{\left(i\right)}$ or $\xi_{r,\mathrm{num}}^{\left(i\right)}$ are almost the same within a 2% deviation, and the next step is to experimentally validate these results.

Because the resonance positions were observed on the *x*- or ξ-axis based on numerical analysis, the oscillation modes on the *y*- or η-axis were measured first (Figure 5).

The amplitudes were maximum on the *x*- or ξ-axis, i.e., y=η=0, indicating that the symmetry was the same as in the numerical analysis. Furthermore, the amplitude and gradient were zero near the fixed boundary, and the experimental devices maintained the fixed boundary condition. Thus, the amplitude distribution demonstrated a fundamental oscillation mode under the fixed-at-both-ends condition of the Euler–Bernoulli theory for the *y*- or η-axis. The fundamental oscillation mode for the *y* direction has been observed in the previous studies [10,17], and it may be suggested that the *y*- or η-axis oscillation mode is dominant in the aspect ratio of the trapezoidal membrane device. Therefore, the oscillation on the *x*- or ξ-axis was measured for various *x*- or ξ positions and input frequencies, as shown in Figure 6 and Figure 7, respectively.

Figure 6 shows that the oscillation modes changed as the input frequency $f_{\mathrm{in}}$ and the resonance positions with the maximum amplitude decreased along the *x*- or ξ-axis as well as in the numerical analysis. Multiple peaks were clearly observed in Figure 7, and they were considered as the *i*th-order (*i* = 1,2,3) resonance frequencies $f_{r,\exp}^{\left(i\right)}$ of each position obtained from the experiment, and the resonance frequencies $f_{r,\exp}^{\left(i\right)}$ and positions $x_{r,\exp}^{\left(i\right)}$ or $\xi_{r,\exp}^{\left(i\right)}$ for the trapezoidal and curved membrane devices, respectively, are shown in Table 2. According to experimental results and numerical analysis, there is almost no difference in the oscillation characteristics of the trapezoidal and curved membrane devices. In this section, the frequency selectivity of both the devices is compared and validated based on the Euler–Bernoulli theory for transverse oscillations of a beam to the fundamental oscillation mode in the *y*- or η direction. The displacement of a 2D plate W(x,y) is typically represented by the product of a function of only *x* and that only of *y* in terms of the method of the separation of variables W(x,y)=X(x)Y(y). However, in the device with a trapezoidal membrane, the *y*-direction width of the membrane is also a function of *x* in this study because its length in the *y*-axis varies along the *x*-axis: Y=Y(x,y). The oscillation of a structure with trapezoidal fixed boundary conditions is a complex interaction of oscillation along the *x*- and *y*-axis, which is difficult to understand and predict.

There are few reports on the oscillation characteristics of clamped trapezoidal membranes, and no rigorous theoretical solution of the equations governing the motion of the trapezoidal plates can be found, while multiple numerical analyses have been reported [41,42]. The bending oscillation of a plate with isotropic mechanical properties can be described in Cartesian coordinates as follows [43]:(3)$$\rho h\frac{\partial^{2}w}{\partial t^{2}}+D\left[\frac{\partial^{4}w}{\partial x^{4}}+2\frac{\partial^{4}w}{\partial x^{2}\partial y^{2}}+\frac{\partial^{4}w}{\partial y^{4}}\right]=p,$$
where *w* is the displacement in the *z* direction, $D=Eh^{3}/\left(12\left(1-\nu^{2}\right)\right)$ is the bending rigidity, *p* is the external pressure, ρ is the mass density, and *h* is the thickness of the plate. Focusing on the resonance position, the oscillation amplitude shows a peak, and the *x*-axis gradient at the resonance position is zero in the case of the trapezoidal device (Figure 6). The section with zero gradient is sufficiently short compared with the *x*-axis length of the trapezoidal membrane *L*, and the local *x*-axis derivative (∂/∂x) at the resonance positions can be considered negligible in Equation (Equation 3). Thus, the *y*-axis oscillation is dominant at the resonance position as follows:(4)$$\rho h\frac{\partial^{2}w}{\partial t^{2}}+D\frac{\partial^{4}w}{\partial y^{4}}=p.$$

By considering the moment of inertia of area in the plate and ν = 0, Equation (Equation 4) coincides with the governing equation of the Euler–Bernoulli beam in the *y* direction [44] as well as our previous estimation [18,24] of the local resonance frequency. The following equation expresses the theoretical resonance frequency $f_{r,\mathrm{theory}}$ for the fundamental oscillation mode with the fixed end conditions: [44]
(5)$$f_{r,\mathrm{theory}}=\frac{22.37}{2\pi l_{y}^{2}\left(x\right)}\sqrt{\frac{Eh^{2}}{12\rho }},$$
where $l_{y}\left(x\right)$ is the length in the *y* direction that continuously changes along the *x* direction. Because *E*, ρ, and *h* are considered to be constant in the xy plane, $f_{r,\mathrm{theory}}$ is primarily determined using $l_{y}\left(x\right)$. Therefore, the resonance frequency decreases along the *x*-axis, which is observed in both the trapezoidal and curved devices. Based on the above equations, our developed theoretical analysis is described below. The following relationship can be deduced from Equation (Equation 5):(6)$$f_{r,\mathrm{theory}}l_{y}^{2}\left(x\right)=C,$$
where $C=22.37\sqrt{\left(Eh^{2}\right)/\left(12\rho \right)}$ is treated as a constant. For the curved membrane device, when the change in the ξ direction at the resonance positions is negligible and the radius of curvature *r* = 30 mm is sufficiently long compared with the membrane width in the η direction, Equations (Equation 5) and (Equation 6) may be applied to the curved device:(7)$$f_{r,\mathrm{theory}}=\frac{22.37}{2\pi l_{\eta }^{2}\left(\xi \right)}\sqrt{\frac{Eh^{2}}{12\rho }},$$
where $l_{\eta }\left(\xi \right)$ is the length in the η direction and
(8)$$f_{r,\mathrm{theory}}l_{\eta }^{2}\left(\xi \right)=C.$$

The Euler–Bernoulli theory can be applied to the case of dξ→0, i.e., a negligible curved angle. The experimental and numerical results are validated using Equations (Equation 5) and (Equation 8). Table 3 presents the values of *C* obtained by substituting the values of $f_{r,\exp}$ and $f_{r,\mathrm{num}}$ into Equations (Equation 5) and (Equation 8) for the trapezoidal and curved devices, respectively.

Notably, the maximum deviation was approximately 2.48%, confirming that *C* is practically constant when the theoretical background governing the motion of the trapezoidal plates is investigated using the simple and essential insight as described above. Furthermore, these values agreed well with each other, and the frequency selectivity of the trapezoidal and curved shape prototypes was quantitatively demonstrated. Both the experimental and numerical results show that the oscillation characteristics of both the devices are quite similar, and the design guidelines obtained in the previous studies [10,17,18] can be used to fabricate an implantable device. Furthermore, this straightened trapezoidal model is extensively used not only in the hearing system research but also in fully insertable cochlear implants, and this model is considered useful for investigating the oscillation properties of membranes with a 3D spiral structure such as the basilar membrane.

### 3.2. Experimental and Numerical Results with Miniaturized Device

Figure 8 shows the natural frequencies $f_{r,\mathrm{num}}^{\left(i\right)}$ and corresponding *i*th-order (*i* = 1, 2, 3) oscillation modes of the miniaturized artificial cochlear sensory epithelium obtained via numerical analysis.

The colormap shows the amplitude of the displacement along the *z*-axis normalized using the maximum value in the ξη plane. The dimensions of the device are determined via numerical analysis based on the experimental device (Figure 2). The results demonstrate that the oscillation amplitude distribution was symmetrical with respect to the ξ-axis. Furthermore, several antinodes were observed as $f_{r,\mathrm{num}}^{\left(i\right)}$ increased, and the maximum amplitude positions, shown in red, migrated along the ξ-axis.

Figure 9 shows the measured amplitude distribution of the miniaturized artificial cochlear sensory epithelium for *i*th-order (*i* = 1, 2, 3) oscillation modes.

For each frequency, the color map describes the amplitude ratio normalized by the maximum value in the plane. Because of the micrometer scale of the device, measurements are difficult and data accuracy is <10%. As the input frequency increases, the resonance positions move along the arc, which is consistent with the numerical analysis result. The frequency responses of the resonance positions $\xi_{r,\exp}^{\left(i\right)}$ were measured (Figure 10).

Multiple peaks were observed, suggesting that they were the resonance frequencies $f_{r,\exp}^{\left(i\right)}$ of each position obtained from the experiment, and the resonance frequencies $f_{r,\exp}^{\left(i\right)}$ and positions $\xi_{r,\exp}^{\left(i\right)}$ are shown in Table 4. Based on Equation (Equation 6), averaged values of $C=6.81\pm 0.0407\;\mathrm{kHz}\;{\mathrm{mm}}^{2}$ and $C=10.4\pm 0.683\;\mathrm{kHz}\;{\mathrm{mm}}^{2}$ were obtained numerically and experimentally, respectively. The numerical and experimental deviations were 0.598% and 6.58%, respectively, indicating that these values are constant. The Euler–Bernoulli beam theory was applied to evaluate the frequency selectivity of the miniaturized artificial cochlear sensory epithelium.

Figure 11 compares the resonance frequencies $f_{r}^{\left(i\right)}$ obtained from the theoretical values obtained from Equation (Equation 7), numerical analysis, and experiment.

The vertical axis represents $f_{r}^{\left(i\right)}$, while the horizontal axis shows the ξ-position. Because $f_{r}^{\left(i\right)}$ is mainly determined by oscillation along the η-axis direction, the Euler–Bernoulli beam was used to predict the frequency selectivity of a film with trapezoidal fixed boundary condition. In all the results, $f_{r}^{\left(i\right)}$ differed as per the ξ position, indicating that the miniaturized device has a frequency selectivity function. Theoretical and experimental results agree well, and the resonance frequency $f_{r}^{\left(i\right)}$ decreases with increasing distance of the ξ position, a characteristic that is quite similar to that of the basilar membrane’s biological hearing system in terms of the frequency selectivity function and tonotopy. This is an important feature for developing a fully insertable cochlear implant, and it enables the prediction of resonance frequency bands in the miniaturized implantable device. The maximum deviation of the resonance positions between the numerical and experimental results was <10% at most, and these values can be considered to be in good agreement. Three pairs of electrodes on the miniature device are used to measure electrical signals generated by the piezoelectric effect and control the oscillations, employing an applied electrical potential. In fact, the electrode positions are determined based on the resonance positions obtained by the numerical analysis. When an external sinusoidal force with the resonance frequency is applied to the device, it oscillates in its natural vibration mode, thus locally increasing the stress at the resonance position. The resonance position is the optimal location for the electrodes because the electricity they generate converts mechanical energy into electrical energy. The results of this study demonstrate that the resonance positions are consistent between the numerical calculations and experiments and the electrodes are installed in the correct positions. However, the maximum deviation between theory and numerical analysis is 33.8%, and the maximum deviation between the numerical and experimental data is 56.4%. When *E* is set to 42.1 GPa, the experimental results agree with the theoretical values, although they differ considerably from those reported in previous studies [39,40]. This is mainly attributed to the device’s layered structure and small size, which can be embedded in the cochlea. The device comprises a 1 μm elastic layer, several nanometer-thick electrodes, and a 1 μm piezoelectric film, which can be considered a composite material, although its physical properties are extremely difficult to predict. Furthermore, the Young’s modulus used in the theoretical and numerical calculations is for the physical properties of bulk PVDF-TrFE and may not be applicable to thin films of the order of $O(10^{-6})$ m. This study focused on the frequency selectivity of the miniaturized device; however, additional investigation of the multilayered device’s physical properties is required.

Experimental and numerical results indicated that the miniaturized device’s resonance frequency band exceeded the human audible range (20 Hz–20 kHz). However, the current experiment was conducted in the air, and the device will be implanted in the cochlea filled with lymphatic fluid in a future clinical application. Because the oscillation characteristics must be evaluated taking into account of the effect of the interaction between the device and the fluid, the resonance frequency band of the device is expected to decrease in the cochlea. In fact, a previous study evaluated the beam array device’s oscillation characteristics in silicone oil, which has similar properties as the lymph fluid, and reported that the frequency band shifted to a lower frequency [24]. Although the resonance frequencies of the miniaturized device exceed the audible frequency range in the air, the frequency range may overlap with the audible range in the cochlea. Because there are still multiple unsolved problems or unknown effects underlying the biological phenomena, the device attributes must be estimated and validated not only from a mechanical engineering viewpoint but also from a biological one in future work.

## 4. Conclusions

In this study, we proposed a novel implantable artificial cochlear sensory epithelium that reproduced the basilar membrane’s frequency selectivity. The following points summarize the main results of this study.

To fabricate a device that can be implanted into the cochlea for the future in vivo testing, the device was curved to fit the cochlea’s spiral structure. The effects of a curved configuration on the resonance characteristics were experimentally and numerically evaluated using an in vitro prototype device. A trapezoidal membrane device developed in previous studies [10,17,18] and a curved one, which was newly fabricated, were prepared, and the resonance frequencies and positions were measured. The resonance characteristics of the trapezoidal and curved membrane devices were in quantitative agreement, suggesting that the effect of curving was negligible and limited to the device fabricated in this study. The frequency selectivity was successfully evaluated via theoretical prediction using the Euler–Bernoulli beam theory. The trapezoidal device is extensively used in the development of fully insertable cochlear implants as well as in auditory system research, demonstrating the model’s validity.The MEMS technology was used to fabricate a miniaturized artificial cochlear sensory epithelium consisting of SU-8 epoxy resin, P(VDF-TrFE) piezoelectric material, and electrodes. Curved trapezoidal fixed boundary conditions were applied to reproduce the basilar membrane’s frequency selectivity. The Ti/Au/Ti patterned electrodes were successfully deposited on the P(VDF-TrFE) film for the future use of the piezoelectric output. The device was designed to be <1 mm, and the curving boundary was shaped to fit the spiral structure that can be implanted in the cochlea.The resonance frequencies and positions of the miniaturized artificial cochlear sensory epithelium were measured to verify the frequency selectivity. Therefore, local resonance frequencies of 157–277 kHz and 103–178 kHz in the air were observed experimentally and numerically, respectively. The resonance positions varied along the longitudinal direction with the resonance frequency, indicating that the frequency selectivity of the basilar membrane was reproduced by the proposed miniaturized device. Although there was a deviation of 50% for resonance frequency between the numerical analysis and experimental results, it was expected that the thin film would alter the physical properties of each layer and additional verification is required. Although the resonance frequency range in the air was above the human audible range (20 Hz–20 kHz), the cochlea is filled with lymphatic fluid, which should shift the frequency range to a lower range.

## Figures and Tables

**Figure 1 micromachines-13-00768-f001:**
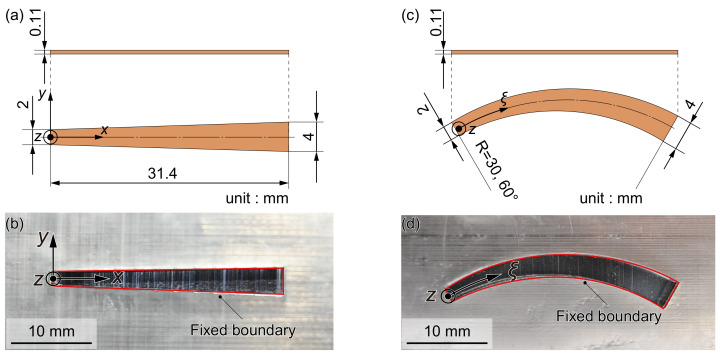
In vitro experiment using the prototype devices of an artificial cochlear sensory epithelium. Trapezoidal and curved membrane devices were fabricated to examine the effect of a curved device on the oscillation characteristics. (**a**) Dimensions of a trapezoidal artificial cochlear sensory epithelium and (**b**) photograph. (**c**) Dimensions of a curved artificial cochlear sensory epithelium and (**d**) photograph. A piezoelectric membrane made of PVDF film was used for these prototype devices. Al thin electrodes were deposited on both sides of the PVDF film. The compound membrane was sandwiched between acrylic plates with a trapezoidal or curved shape hole, and fixed boundaries of the prototype devices were determined by the edges of the hole. In the photographs (**b**,**d**), the red solid lines indicate the fixed boundaries of the devices, silver color the acrylic plate with the trapezoidal or curved shape hole, and black color the Al electrodes deposited on the PVDF film.

**Figure 2 micromachines-13-00768-f002:**
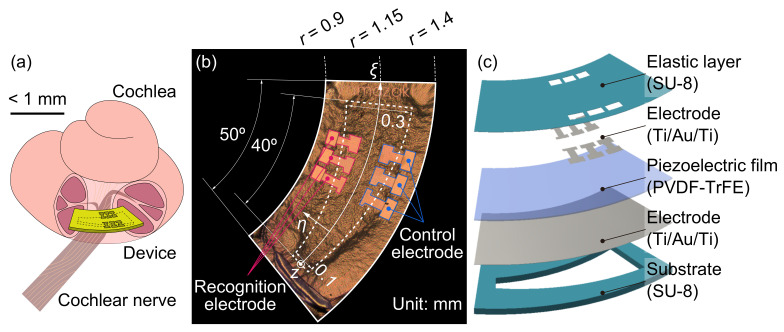
(**a**) Schematic of a miniaturized artificial cochlear sensory epithelium for in vivo experiments. The device is designed to be implanted in the cochlea to develop a fully insertable cochlear implant. (**b**) Photomicrograph of the miniaturized device. Recognition electrodes for measuring the piezoelectric output and control electrodes for applying electrical potential to amplify oscillations are deposited with reference to our previous researches [10,17,18]. (**c**) Configuration of the device. The piezoelectric effect generates potential differences that are used to electrically stimulate the auditory nerve in an attempt to restore hearing.

**Figure 3 micromachines-13-00768-f003:**
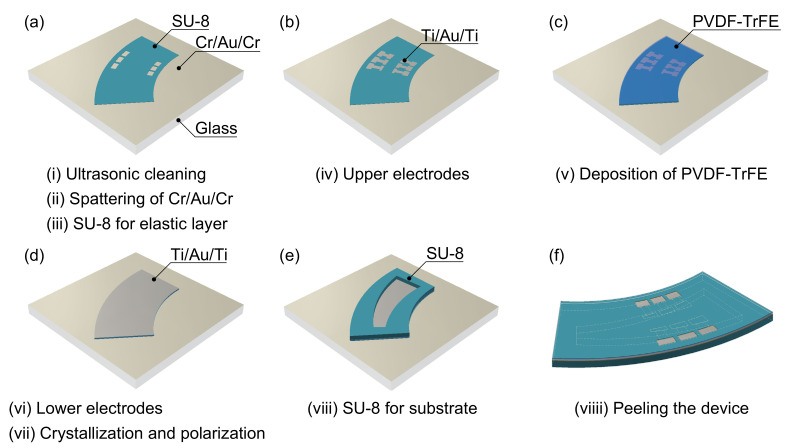
Technique of fabricating a miniaturized artificial cochlear sensory epithelium for implantation in guinea pigs. (**a**) A glass substrate is ultrasonically cleaned using isopropanol. Cr/Au/Cr as a sacrificial layer is sputtered on the glass substrate. On the surface of Cr, an elastic layer of SU-8 is deposited. (**b**) Using the lift-off method, the Au for the upper electrodes is deposited on SU-8 with Ti as an adhesion layer. (**c**) On the surface of the SU-8 and Au electrodes, a PVDF-TrFE film is deposited. (**d**) Using the lift-off method, the Au for the lower electrodes is deposited on PVDF-TrFE with Ti as an adhesion layer. (**e**) SU-8 is deposited on the lower electrodes of the substrate with a curved–trapezoidal aperture. (**f**) A Cr etchant is used to peel the miniaturized artificial cochlear sensory epithelium from the glass substrate.

**Figure 4 micromachines-13-00768-f004:**
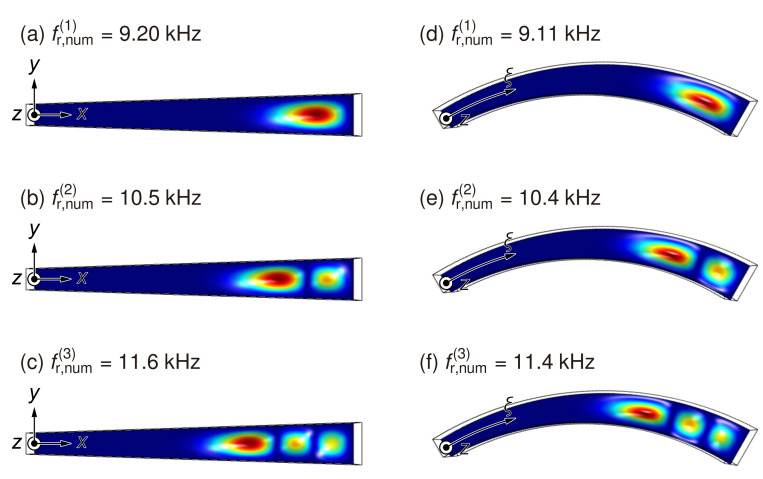
Numerical results of amplitude distribution in the xy plane for (**a**–**c**) the trapezoidal and (**d**–**f**) curved membrane device for various $f_{r,\mathrm{num}}^{\left(i\right)}$, where *i* denotes the oscillation mode. Contours show the relative amplitude of each natural frequency, normalized by the maximum value of the oscillation amplitude. The amplitude distribution is symmetrical about the *x*- or ξ-axis, and the fundamental oscillation mode is in the direction of the *y*- or η-axis. Resonance positions indicated by the maximum amplitude are obtained for both the devices. Each device has the different resonance positions depending on the normal modes, and it is determined that both the devices have frequency selectivity. When the trapezoidal and curved devices are compared, it is confirmed that there is a negligible difference in their natural frequencies or eigenmodes.

**Figure 5 micromachines-13-00768-f005:**
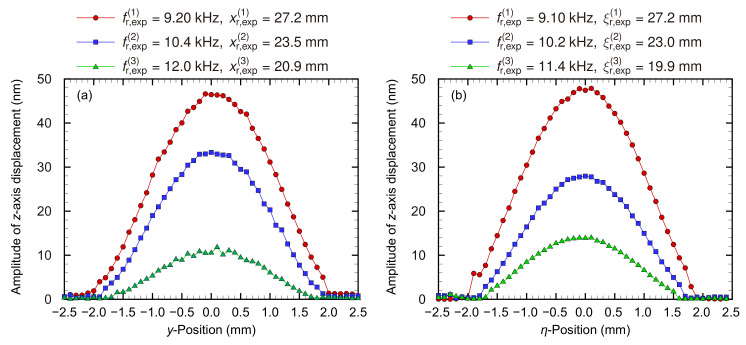
Experimentally obtained relationship between the oscillation amplitude and *y*-position of (**a**) the trapezoidal or (**b**) curved membrane device, where the oscillation is induced with each resonance frequency at the resonance *y*-positions. The amplitude distribution is symmetrical at y=η=0 mm, and the amplitude is maximum at y=η=0 mm. Furthermore, the amplitude and slope are zero near the fixed boundary. Thus, the amplitude distribution shows a fundamental oscillation mode for the *y* and η axes under the fixed-at-both-ends condition of the Euler–Bernoulli theory.

**Figure 6 micromachines-13-00768-f006:**
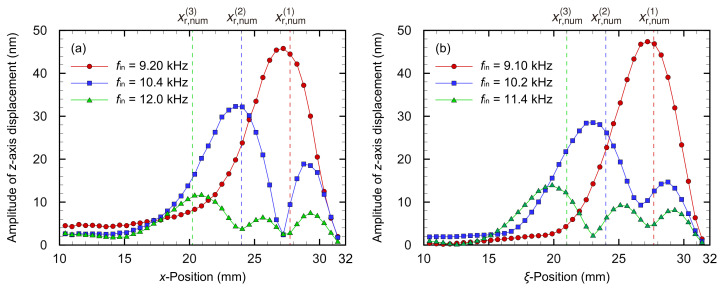
Experimentally obtained oscillation amplitude distribution in the *x*-axis direction of (**a**) the trapezoidal and (**b**) curved membrane devices, where the oscillation is induced with each resonance frequency at the resonance positions. The dashed line indicates the resonance positions in the numerical analysis. For each eigenmode, the positions where the amplitude is maximum, i.e., the resonance positions, are observed, and the results correspond well with the numerical analysis. The amplitude distributions for the trapezoidal and curved devices are in good agreement, suggesting that the curving device has a small effect on the oscillation characteristics of the trapezoidal membrane device.

**Figure 7 micromachines-13-00768-f007:**
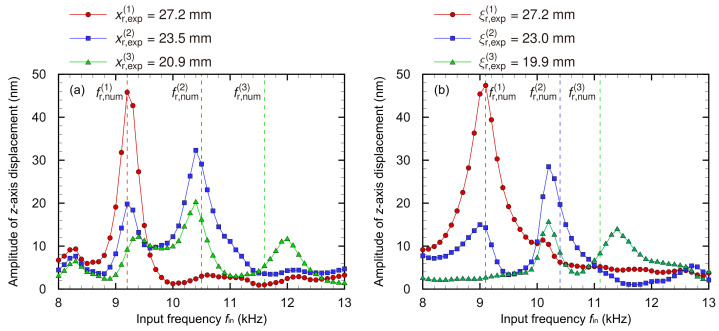
Experimentally obtained resonance characteristics at each resonance position of (**a**) the trapezoidal and (**b**) curved membrane devices. A frequency is observed at which the amplitude is maximized, and this frequency is defined as the resonance frequency based on the experimental results. The dashed line indicates the resonance frequencies obtained via the numerical analysis. Both devices’ experimental and numerical results are in good agreement. Table 1 and Table 2 indicate that the resonance frequencies for the trapezoidal and curved membrane devices are almost the same.

**Figure 8 micromachines-13-00768-f008:**
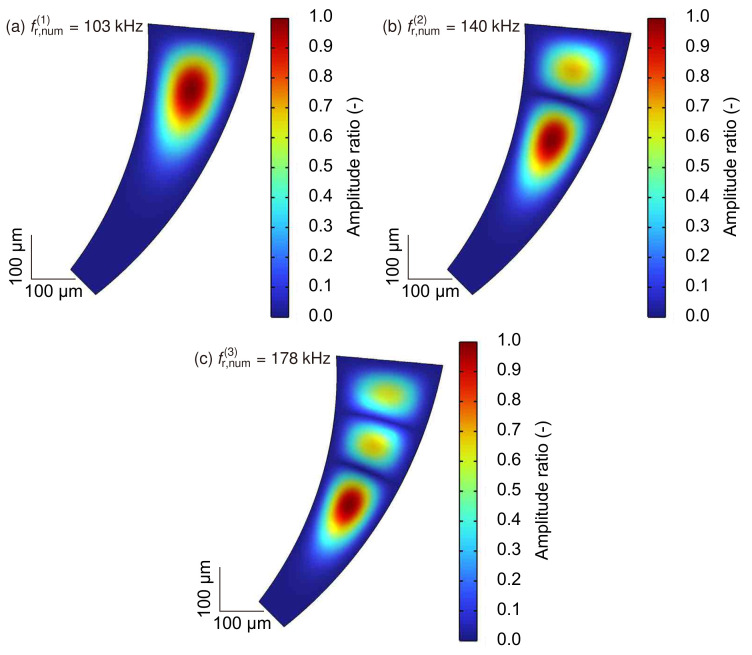
Numerical analysis of amplitude distribution in the xy plane for the miniaturized artificial cochlear sensory epithelium. Contours demonstrate the relative amplitude normalized by the maximum value of the oscillation amplitude for the natural frequency (**a**–**c**). Resonance positions indicated by the maximum amplitude are obtained for all eigenmodes. The device has different resonance positions based on the normal modes, and it is concluded that the miniaturized device has the function of frequency selectivity.

**Figure 9 micromachines-13-00768-f009:**

Experimental results of amplitude distribution in the xy plane for the miniaturized artificial cochlear sensory epithelium. Contours show the relative amplitude normalized by the maximum value of the oscillation amplitude when the driving force with the frequency of (**a**–**c**) was applied. Resonance positions indicated by the maximum amplitude are obtained for every resonance frequency. The device exhibits different resonance positions depending on the normal modes, and this tendency is observed in the numerical analysis.

**Figure 10 micromachines-13-00768-f010:**
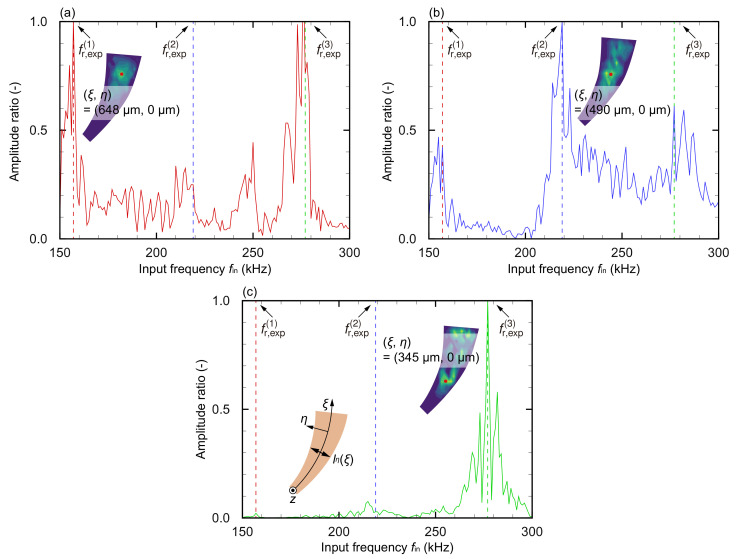
Experimentally obtained resonance characteristics of the miniaturized artificial cochlear sensory epithelium at (**a**) (ξ,η) = (648 μm, 0 μm), (**b**) (ξ,η) = (490 μm, μ0 m), and (**c**) (ξ,η) = (345 μm, 0 μm). The red dots indicate the measurement points defined by the curvilinear coordinates. Several frequencies are observed where the amplitude is maximized, and these frequencies are defined as the resonance frequency based on the experimental results. Therefore, each measurement position has the local resonance frequency, and it is demonstrated that the miniaturized device reflects the frequency selectivity function.

**Figure 11 micromachines-13-00768-f011:**
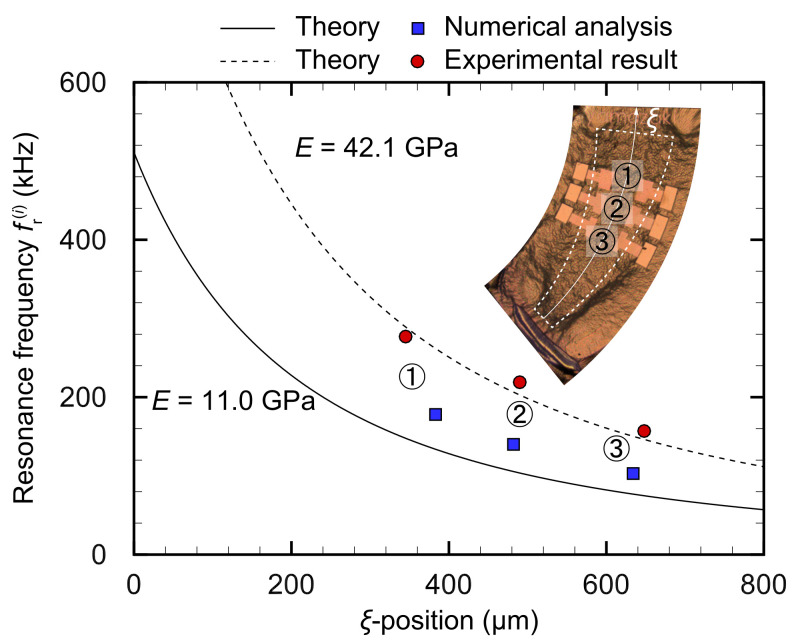
Comparisons of resonance frequencies obtained from theoretical, numerical, and experimental results in a miniaturized artificial cochlear sensory epithelium for in vivo testing. Because the resonance frequencies may be mainly determined by the oscillation of the η-axis, theoretical values are calculated using Euler–Bernoulli beams. Although there is a maximum deviation of 33.8% between theory and numerical calculation and of 56.4% between the experimental and numerical results, the frequency selectivity performance has been evaluated. We believe that Young’s modulus is the main cause of the deviation, and it is necessary to evaluate the physical properties of thin films on a micrometer scale.

**Table 1 micromachines-13-00768-t001:** Natural frequencies $f_{r,\mathrm{num}}^{\left(i\right)}$ and resonance positions $x_{r,\mathrm{num}}^{\left(i\right)}$ or $\xi_{r,\mathrm{num}}^{\left(i\right)}$ for the trapezoidal and curved membrane devices based on the numerical analysis. The maximum deviation of $f_{r,\mathrm{num}}^{\left(i\right)}$ between the trapezoidal and curved membrane devices is 1.75% and that between $x_{r,\mathrm{num}}^{\left(i\right)}$ and $\xi_{r,\mathrm{num}}^{\left(i\right)}$ is 0.476%. Therefore, it is possible to conclude that these results are quantitatively consistent.

	Trapezoidal		Curved	
i	$f_{r,\mathrm{num}}^{\left(i\right)}$ **(kHz)**	$x_{r,\mathrm{num}}^{\left(i\right)}$ **(mm)**	$f_{r,\mathrm{num}}^{\left(i\right)}$ **(kHz)**	$\xi_{r,\mathrm{num}}^{\left(i\right)}$ **(mm)**
1	9.20	27.6	9.11	27.6
2	10.5	24.0	10.4	24.0
3	11.6	21.1	11.4	21.0

**Table 2 micromachines-13-00768-t002:** Resonance frequencies $f_{r,\exp}^{\left(i\right)}$ and resonance positions $x_{r,\exp}^{\left(i\right)}$ or $\xi_{r,\exp}^{\left(i\right)}$ for the trapezoidal and curved prototype devices based on the experiment. The maximum deviation of $f_{r,\exp}^{\left(i\right)}$ between the trapezoidal and curved membrane devices is 5.26% and that between $x_{r,\exp}^{\left(i\right)}$ and $\xi_{r,\exp}^{\left(i\right)}$ is 5.03%. Therefore, the oscillation characteristics of the trapezoidal and curved membrane devices fabricated in this study are in good agreement.

	Trapezoidal		Curved	
i	$f_{r,\exp}^{\left(i\right)}$ **(kHz)**	$x_{r,\exp}^{\left(i\right)}$ **(mm)**	$f_{r,\exp}^{\left(i\right)}$ **(kHz)**	$\xi_{r,\exp}^{\left(i\right)}$ **(mm)**
1	9.20	27.2	9.10	27.2
2	10.4	23.5	10.2	23.0
3	12.0	20.9	11.4	19.9

**Table 3 micromachines-13-00768-t003:** Constant values *C* in Equations (Equation 5) and (Equation 8) derived from the Euler–Bernoulli theory. They can all be considered constant, and the theoretical predictions of frequency selectivity are clearly confirmed.

	*C* (kHz mm${}^{2}$)			
	**Trapezoidal**		**Curved**	
i	**Numerical**	**Experimental**	**Numerical**	**Experimental**
1	130	128	129	127
2	131	127	129	122
3	130	133	127	127
Ave.	130 ± 0.54	130 ± 3.21	127 ± 1.28	125 ± 2.42

**Table 4 micromachines-13-00768-t004:** Resonance frequency ($f_{r,\mathrm{num}}^{\left(i\right)}$ and $f_{r,\exp}^{\left(i\right)}$) and resonance positions ($\xi_{r,\mathrm{num}}^{\left(i\right)}$ and $\xi_{r,\exp}^{\left(i\right)}$) of the MEMS device based on the experimental and numerical results. Although the maximum deviation between $f_{r,\mathrm{num}}^{\left(i\right)}$ and $f_{r,\exp}^{\left(i\right)}$ is 56.4%, the maximum deviation between $\xi_{r,\mathrm{num}}^{\left(i\right)}$ and $\xi_{r,\exp}^{\left(i\right)}$ is 9.92% and the oscillation modes are predictable for the optimization of the electrode positions.

* **i** *	$f_{r,\mathrm{num}}^{\left(i\right)}$ (kHz)	$\xi_{r,\mathrm{num}}^{\left(i\right)}$ (μm)	$f_{r,\exp}^{\left(i\right)}$ (kHz)	$\xi_{r,\exp}^{\left(i\right)}$ (μm)
1	103	634	157	648
2	140	482	219	490
3	178	383	277	345

## Data Availability

The data are available from the corresponding author on reasonable request.

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
