# Peer review of "Oscillation Characteristics of an Artificial Cochlear Sensory Epithelium Optimized for a Micrometer-Scale Curved Structure"

_micromachines, 2022, doi:10.3390/mi13050768_

Round 1

Reviewer 1 Report

In this manuscript, the authors proposed miniaturized artificial cochlear sensory epithelium for future implantation tests. Since the authors achieved fabrication of the artificial cochlear sensory epithelium and analyses of its oscillation characteristics using experimental and numerical methods, I recommend publication of the manuscript on Micromachines. However, some details are unclear in the manuscript. My concerns and questions are noted below.

1.

In section 2.1, Line 6, the authors described “a PVDF membrane with a thickness of 110 um that was deposited Al was fixed to the jig.”. However, it is difficult to understand exactly what kind of structure it is. Does the sentence mean that the jig is a plate-like object with a hole and the membrane is sandwiched between two jigs? I recommend that the authors describe it accurately.

Also, it is difficult to understand Figure1. Does it mean that the membrane is sandwiched between the jigs and the membrane is observable through the hole in the jig? In addition, I think it would be better to clearly indicate what the silver or black parts are.

2.

Starting with Tables 1 and 2, the order in which the figures and tables appear is out of order or placed far from the text. It would be better to consider their placement to improve readability.

3.

The authors show equation 1-8. However, I cannot figure out which are the authors' original equations. If they are not original, it would be better to cite references. If they are original, I think there is too little explanation of the equation derivation, so I recommend the authors add assumptions of these equations for easy to understand them.

Reviewer 2 Report

COMMENTS TO AUTHOR:

This work reports a novel implantable artificial cochlear sensory epithelium that reproduces

the basilar membrane’s frequency selectivity. This work is well designed and the results is of guiding significance in future research of artificial cochlear. I suggest publication of this manuscript in Micromachines after minor revisions.

  1. In this work, did you compare the performan ofartificial cochlear made of PVDF and PVDF-TrFE? Can you do some comparative analysis.
  2. Dose the piezoelectricproperty of Organic piezoelectric materials has significant effect on the performance of artificial cochlear, if so, related characterizations should be provided.
